# Anhydrous Proton Conductivity in HAp-Collagen Composite

**Tomoki Furuseki * and Yasumitsu Matsuo ***

Faculty of Science & Engineering, Setsunan University, Ikeda-Nakamachi, Neyagawa 572-8508, Japan
* Correspondence: 20d903ft@edu.setsunan.ac.jp (T.F.); ymatsuo@lif.setsunan.ac.jp (Y.M.)

**Abstract:** It is well known that a proton conductor is needed as an electrolyte of hydrogen fuel cells, which are attracting attention as an environmentally friendly next-generation device. In particular, anhydrous proton-conducting electrolytes are highly desired because of their advantages, such as high catalytic efficiency and the ability to operate at high temperatures, which will lead to the further development of fuel cells. In this study, we have investigated the proton-conducting properties of the hydroxyapatite (HAp)-collagen composite without external humidification conditions. It was found that, by injecting HAp into collagen, the electrical conductivity becomes higher than that of the HAp or the collagen. Moreover, the motional narrowing of the proton NMR line is observed above 130 °C. These results indicate that the electrical conductivity observed in the HAp-collagen composite is caused by mobile protons. Furthermore, we measured the proton conduction of HAp-collagen composite films with different HAp contents and investigated the necessity of the appearance of proton conductivity in HAp-collagen composites. HAp content ($n = 0$–0.38) is the number of HAp per collagen peptide representing Gly-Pro-Hyp. These results indicate that injection of HAp into collagen decreases the activation energy of proton conduction which becomes almost constant above a HAp content $n$ of 0.3. It is deduced that the proton-conduction pathway in the HAp-collagen composite is fully formed above $n = 0.3$. Furthermore, these results indicate that the value of the activation energy of proton conductivity was lowered, accompanied by the formation of the HAp-collagen composite, and saturated at $n > 0.3$. From these results, the HAp-collagen composite forms the proton-conduction pathway $n > 0.3$ and becomes the proton conductor with no external humidification in the condition of $n > 0.3$ above 130 °C.

**Keywords:** biomaterials; proton conductivity; electrolyte; HAp; collagen

## 1. Introduction

In recent years, the investigation into fuel cells, to find environmentally friendly next-generation energy sources, has been very active. Proton conductors, which make up the electrolyte, a significant part of the fuel cell, have been investigated with much interest [1]. In particular, solid proton conductors are used extensively as fuel cell electrolytes due to their portability and stability [2–5]. Moreover, it is known that solid proton conductors are used not only as the electrolyte of fuel cells but also in the hydrogen sensor as a substitute for the semiconductor [6–13]. Recently, many studies have been carried out on the proton conductor of biopolymers such as collagen, chitin, and cellulose [6–13]. These biopolymers are found in many discarded materials, such as crab and shrimp shells, fish scales, and tree fragments, and can be obtained in large quantities and at a low cost. Furthermore, biopolymers are decomposed naturally. Therefore, biopolymers are attracting attention as materials that are not environmentally detrimental. In addition to these benefits, biopolymers are known to be proton conductors under high humidity conditions. For example, collagen, a fibrous protein, causes proton conduction due to the formation of water cross-links in the glycine-proline-hydroxyproline sequence that is abundant in its structure. Therefore, from zero humidity to 60% relative humidity, there is no proton conduction, but above 60% relative humidity proton conductivity increases with increasing humidity. There are many investigations into the mechanism of proton conductivity in biopolymers, and

proton conductivity is caused by the formation of cross-links between the biopolymer main chain and water molecules, causing the proton-conduction pathway to appear [6–10,14,15]. Studies of the proton conductivity mechanism and the fuel cell electrolyte using these biopolymers have already been carried out with much interest. It is also known that there are some disadvantages of using humidified proton conductors, which transport protons using water. For example, the difficulties of operating above 100 °C, where catalytic efficiency increases, and of operating below 0 °C, when humidifying instruments are needed. For this reason, the development of materials with proton conductivity under high-temperature conditions without external humidification conditions is strongly desired. It is known that inorganic materials such as $CsHXO_4$, $M_3H(XO_4)_2$, and $CsH_2PO_4$ and organic materials composed of imidazole and phosphoric acid become proton conductors without external humidification conditions [16–22]. These non-humidified proton conductors have been shown to transport protons by utilizing the breaking and rearrangement of the hydrogen bond or the mobile amino groups in their structures, and therefore these functions are realized above 100 °C. Hydroxyapatite (HAp) is also known as one of the materials that can cause proton conduction under high-temperature conditions of around 800 °C [23–26]. HAp is one of the calcium phosphate compounds, which can be easily chemically synthesized by mixing substances containing calcium and phosphate ions. In addition, HAp is known to be easily prepared in crystalline form, and HAp crystals can be obtained using wet or dry methods [27–31]. Furthermore, HAp is known to exist in the bodies of many organisms, and bones and teeth are composed mainly of HAp. HAp is responsible for increasing bone strength by forming a composite with collagen, known as a fibrous protein [32]. It is known that the HAp and collagen composite are formed by the chelation of calcium in HAp to the amide bond-derived carboxyl oxygen in collagen [33,34]. Thus, HAp is known to have a high affinity for collagen molecules. The crystals of HAp are hexagonal systems with a space group $P6_3/m$, and the orientation of hydroxyl groups in the c-axis direction [35]. M Yashima et al. reported that proton migration using hydroxyl groups on the c-axis of HAp occurs in proton conduction [23]. Thus, HAp has the potential to be applied as a non-humidity proton conductor. However, the proton conductivity of HAp at around 200 °C is negligibly low because proton conductivity appears from 700 to 900 °C. It is known that HAp in vivo mainly bonds with collagen fiber and forms a structure in the collagen fiber which is in parallel with the c-axis of the HAp crystal [36]. This structure gives the HAp-collagen composite its strong and flexible properties. This flexible structure has the potential to yield higher proton conductivity compared with HAp. However, to date, there has been no investigation of the anhydrous proton conductivity in HAp-collagen composite. In the present study, we have investigated the proton conductivity at around 130 °C in HAp-collagen composite films without external humidification conditions. These results will be helpful in the development of new environmentally friendly, non-humidified proton conductors composed of biomaterials.

## 2. Materials and Methods

### 2.1. Preparation of HAp Film, Collagen Film, and HAp-Collagen Composite Film

The HAp-collagen composite films were prepared from tilapia fish scales (Nitta gelatin Inc. Osaka, Japan). The collagen film was prepared using purified HAp-collagen composite film. The collagen film was obtained by decalcifying purchased scales (HAp-collagen composite) in 0.1M EDTA (Nacarai tesque, Osaka, Japan) for 48 h, removing all minerals, and then washing with 99% acetone (Nacarai tesque, Osaka, Japan) and Milli-Q water. The yield of purified collagen was about 55% by weight ratio. Before the measurements, we confirmed using FT-IR spectra that all HAp molecules in collagen films had been removed. Figure 1 shows photographs of the HAp-collagen composite film and the collagen film. In Figure 1, the purified collagen film is translucent (Figure 1b), although the HAp-collagen composite film is white (Figure 1a). The thickness of the specimen was adjusted to be 0.10 mm. In addition, specimens were cut to 1.0 cm × 0.70 cm and used in measurements.

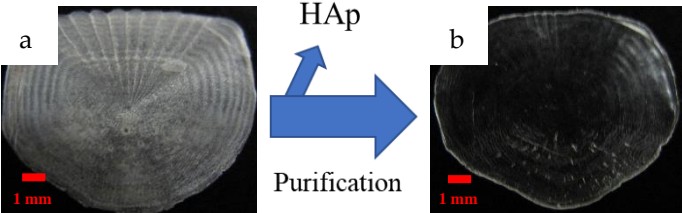

HAp-collagen composite film          Extracted collagen film

**Figure 1.** Hap-collagen composite film and collagen film. (**a**) HAp-collagen composite film from fish. HAp-collagen composite film from fish. (**b**) Purified collagen film.

Figure 2 shows micrographs of the collagen film and the HAp-collagen composite film at 25 to 200 °C. Figure 2a–c show collagen film, and Figure 2d–g shows HAp-collagen composite film. In the collagen film, the unique pattern of the scale was visible up to 150 °C, but no longer visible at 160 °C. These results indicate that collagen softens at temperatures above 160 °C. However, the HAp-collagen composite film showed no softening at 200 °C. These results indicate that thermal durability is improved by the formation of HAp-collagen composite films.

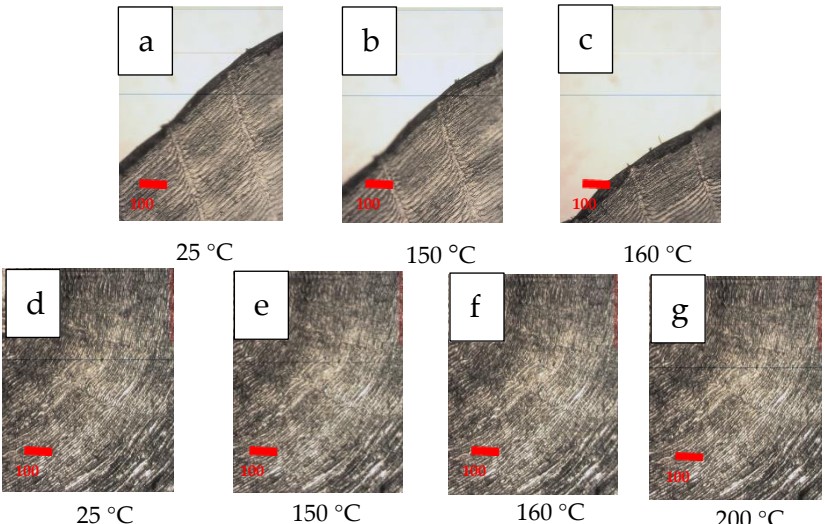

**Figure 2.** Temperature-dependent changes in collagen films and HAp-collagen composite films. Pictures (**a**–**c**) are extracted collagen films. Pictures (**d**–**g**) are HAp 40% HAp-collagen composite films synthesized using Method 2-2. In collagen images (**a**,**b**), there is still no change in the scale pattern and shape, but in (**c**), the pattern and shape have changed due to softening. The HAp-collagen composite film images (**d**–**g**) showed no change in scale pattern and shape.

In the present work, HAp films were prepared using purified HAp (HAp-100, Taiheik-agaku, Osaka, Japan). HAp films were grown by mixing 2.50 g of HAp100, 0.125 g of citric acid (Nacarai tesque, Osaka, Japan), 50.0 g of zirconia beads (Taiheikagaku, Osaka, Japan), and 20.0 g of ethanol (Nacarai tesque, Osaka, Japan), which was then stirred for 8 h and then shaped and sintered at 1000 °C.

### 2.2. Synthesis of HAp-Collagen Composites

Figure 3 shows the schematic diagram of the preparation method for HAp-collagen composite. As shown in Figure 3, HAp-collagen composite films with different HAp weight ratios were prepared by synthesizing HAp within collagen films. The HAp-collagen composite films were synthesized using a wet method using purified collagen film as a scaffold to introduce HAp. $Na_2HPO_4$ (Nacarai tesque, Osaka, Japan) and $CaCl_2$ (Nacarai tesque, Osaka, Japan) were used for the synthesis of HAp. Milli-Q water was used as the

solvent. The purified collagen film was agitated with 200 mM $CaCl_2$ to inject calcium ions into the collagen, and the free ions that were not injected were washed out by Milli-Q water. The HAp-collagen composite film was then prepared by agitating the calcium ion-loaded collagen films with 120 mM disodium hydrogen phosphate solution to synthesize HAp within the collagen. Finally, the HAp-collagen composite films produced were washed with Milli-Q water to remove unreacted disodium hydrogen phosphate. This series of operations was repeated as one cycle to produce collagen composites with prepared weight concentrations of HAp. This cycle was performed for 10~100 cycles.

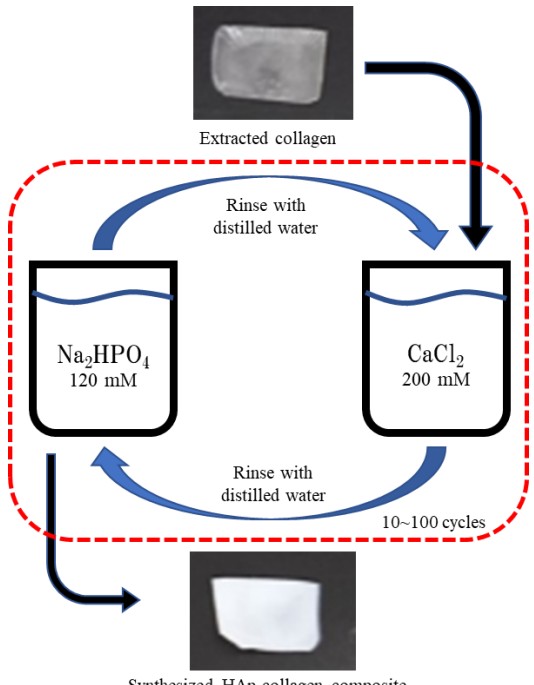

**Figure 3.** Method for synthesis of HAp-collagen composite films.

Figure 4 shows a photograph of HAp-collagen composite film specimens in which HAp was synthesized in the structure using the wet synthesis method. We synthesized HAp-collagen composite films with HAp weight ratios of 5, 15, 25, 30 and 40% from HAp with calcium oxide and disodium hydrogen phosphate. In Figure 4, the samples can be seen to become whiter as the HAp weight ratio increases.

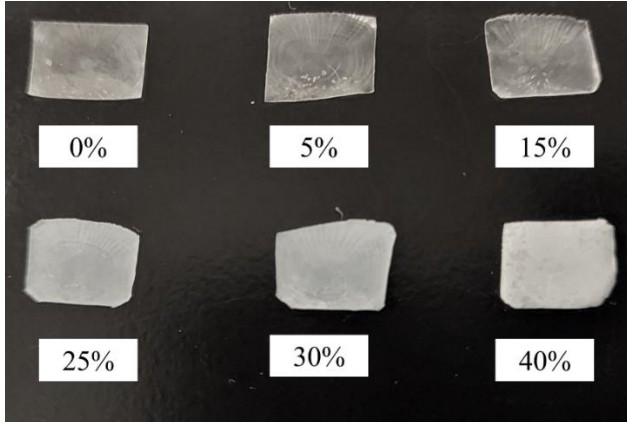

**Figure 4.** HAp-collagen composite films for each HAp weight ratio.

### 2.3. Measurement of FT-IR

FT-IR spectra of the prepared collagen film and the HAp-collagen composite film were measured in the wavenumber range of 400–2000 cm$^{-1}$ using an FT-IR spectrometer (iS-5, Thermo Fisher Scientific, Waltham, MA, USA).

### 2.4. Impedance Measurements

The impedance measurements of the HAp film, the collagen film, and the HAp-collagen composite film were carried out using an LCR meter (E4980A: Agilent technology, Santa Clara, CA, USA). The impedance was obtained using frequencies from 1 kHz to 1 MHz in a temperature range between 100 and 200 °C. The impedance of the collagen film and HAp-collagen composite film were measured both in the direction parallel to the collagen fiber and in the direction perpendicular to the collagen fiber. The specimens used for the impedance measurement were put in a dry vacuum environment of 5.33 KPa. Gold electrodes were deposited on the specimens, as shown in Figure 5.

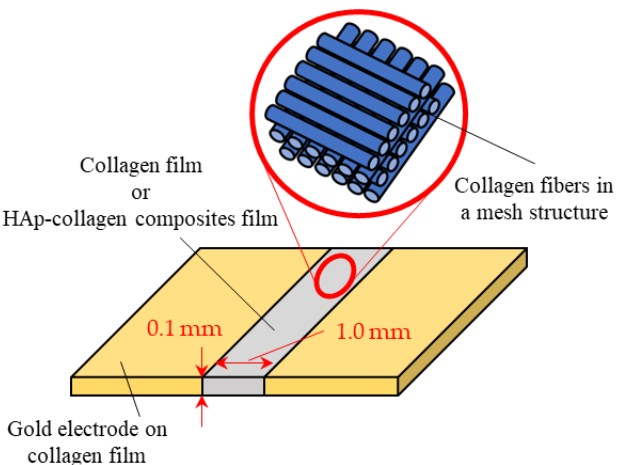

**Figure 5.** Schematic diagram of impedance measurement.

### 2.5. Data Fitting

We performed data fitting to analyze the data obtained from the impedance measurements. The DC conduction $\sigma_0$ was obtained from the following Equation (1):

$$\sigma_{\text{AC}} = \sigma_0 + \omega\varepsilon_0\varepsilon'' \tag{1}$$

Equation (1) is obtained assuming that the equivalent circuit of the sample is described by a simple parallel circuit of capacitance and resistance. Here, $\varepsilon''$ and $\varepsilon_0$ are the imaginary part of the dielectric constant and dielectric constant in a vacuum, respectively. The symbol $\omega$ is the angular frequency, and $\sigma_0$ is DC electrical conductivity, which is calculated from the real part of the impedance. In the case that $\varepsilon''$ is independent of the frequency, $\sigma_{\text{AC}}$ is described by the simple parallel equivalent circuit of the capacitance and resistance and monotonously increases with increasing frequency $f$ (= $\omega/2\pi$).

### 2.6. Measurement of $^1$H-NMR

$^1$H-NMR measurements of HAp-collagen composite films were performed using a home-made solid-state NMR spectrometer, which consists of a Pulse Generator (N146-4746AM, Thamway, Shizuoka, Japan), a 300 Hz Multi-Function Generator (WAVE FACTORY, Kanagawa, Japan), a Multi-Function Synthesizer (WAVE FACTORY, Kanagawa, Japan), and an amplifier (Thamway, Shizuoka, Japan) for the pulse transmitter and receiver (Thamway, Shizuoka, Japan). The specimens were sealed in a 5 mm diameter glass tube. The $^1$H-NMR absorption lines were observed at a resonance frequency of 9.979 MHz in the temperature range of 120–180 °C.

## 3. Results and Discussion

Figure 6a–c shows the FT-IR spectra in the HAp film, the collagen film, and the HAp-collagen composite film, respectively. In Figure 6c, the HAp-collagen composite film was prepared by injecting HAp into the collagen film with a HAp content of 0.38 per collagen molecule (Gly-Pro-Hyp). As shown in Figure 6b, the absorption lines are observed at 1650 $cm^{-1}$, 1540 $cm^{-1}$, 1480 $cm^{-1}$, and 1240 $cm^{-1}$ in the collagen film, respectively. It is known that these absorptions at 1650 $cm^{-1}$, 1540 $cm^{-1}$, 1480 $cm^{-1}$, and 1240 $cm^{-1}$ are those of amide I, amide II, and amide III derived from collagen, and these are identified as C=O stretching vibration; N-H vending and C-N stretching vibrations; N-H stretching; C-N stretching and N-H vending, respectively [37,38]. In this way, the FT-IR spectrum in Figure 6b is consistent with that of collagen. On the other hand, by injecting HAp, sharp absorption peaks appear at 500–600 $cm^{-1}$ and 1000–1100 $cm^{-1}$. These peaks are consistent with the FT-IR spectrum of HAp in Figure 6a. It is also known that HAp-specific P-O-derived absorption bands were observed at 560 $cm^{-1}$, 600 $cm^{-1}$, and 1030 $cm^{-1}$ [39]. These results indicate that HAp-collagen composite film is synthesized. In this way, we can confirm the existence of the HAp in the collagen film from the IR spectra.

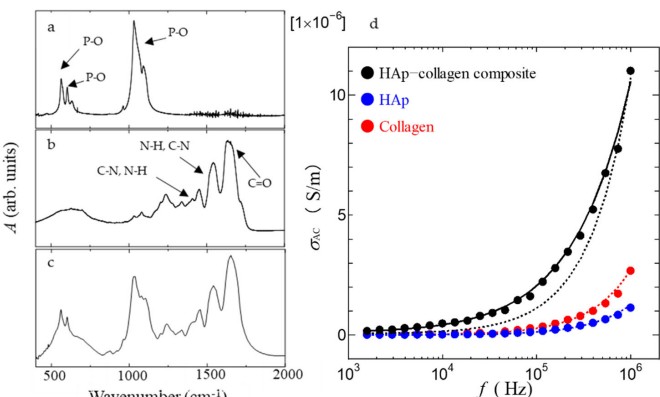

**Figure 6.** FT-IR spectra and impedance characteristics of HAp-collagen composite film. FT-IR spectra of (**a**) HAp film, (**b**) collagen film, and (**c**) HAp-collagen composite film. The graph in (**d**) is the frequency dependence of $\sigma_{AC}$ in the collagen film, the HAp film, and the HAp-collagen composite film at 150 °C. Dotted lines show the results fitted by $\sigma_{AC} = \sigma_0 + \omega\varepsilon_0\varepsilon''$. The solid line shows the results calculated by Equation (3).

Figure 6d shows the frequency dependence of the electrical conductivity of dry collagen films, HAp films, and HAp-collagen composite films at 150 °C. Figure 6d shows that the HAp-collagen composite films exhibit higher conductivity than the collagen films and HAp films in all frequency regions. These results indicate that introduction of HAp films into collagen films results in higher conductivity. In order to investigate the origin of the AC electrical conductivity of HAp-collagen composite films in detail, Equation (1) was used to analyze the frequency dependence of the AC electrical conductivity of these materials. The calculated values are shown by the dotted lines in Figure 6d. Figure 6d shows that the calculated curves (dotted lines) for the collagen films and HAp films agree well with the measured values. This result indicates that the equivalent circuits of the collagen film and the HAp film are described by a parallel circuit of capacitance and resistance. Therefore, we can obtain the DC conductivity of $\sigma_0$ from the dotted line, and the DC electrical conductivities of collagen film and HAp film are estimated to be $9.00 \times 10^{-9}$ S/m and $5.00 \times 10^{-10}$ S/m. In this way, these electrical conductivities are significantly low. On the other hand, as shown by the dotted line of the HAp-collagen composite film in Figure 6d, the frequency dependence of the HAp-collagen composite film cannot be fitted by the simple equation $\sigma_{AC} = \sigma_0 + \omega\varepsilon_0\varepsilon''$. That is, $\varepsilon''$, strongly depends on the frequency, and the equivalent circuit of the HAp-collagen composite film cannot be described by a simple parallel circuit of capacitance and resistance. It is well known that a polymer with bonding

between the polymer chain and polar molecule shows dielectric dispersion around 100 kHz. Therefore, this result indicates that dielectric dispersion appears by introducing HAp into collagen film. That is, by forming the HAp-collagen composite film, the mobile molecules which respond to the electric field appear. In addition, the OH groups in the HAp-collagen composite films are influential in mobile molecules. Considering that the electrical conductivity also increases simultaneously with the appearance of mobile molecules by the formation of the HAp-collagen composite film, it is speculated that the increase in electrical conductivity is caused by proton motion. To investigate the proton dynamics directly, we carried out [1]H-NMR analysis.

Figure 7a shows the NMR absorption lines of the HAp-collagen composite film at 130, 160 and 180 °C. In Figure 7a, we can observe the narrowing of the NMR absorption lines in the HAp-collagen composite film with the increase in temperature. This result indicates that the motional narrowing of the NMR line width due to proton migration is observed in HAp-collagen composite film.

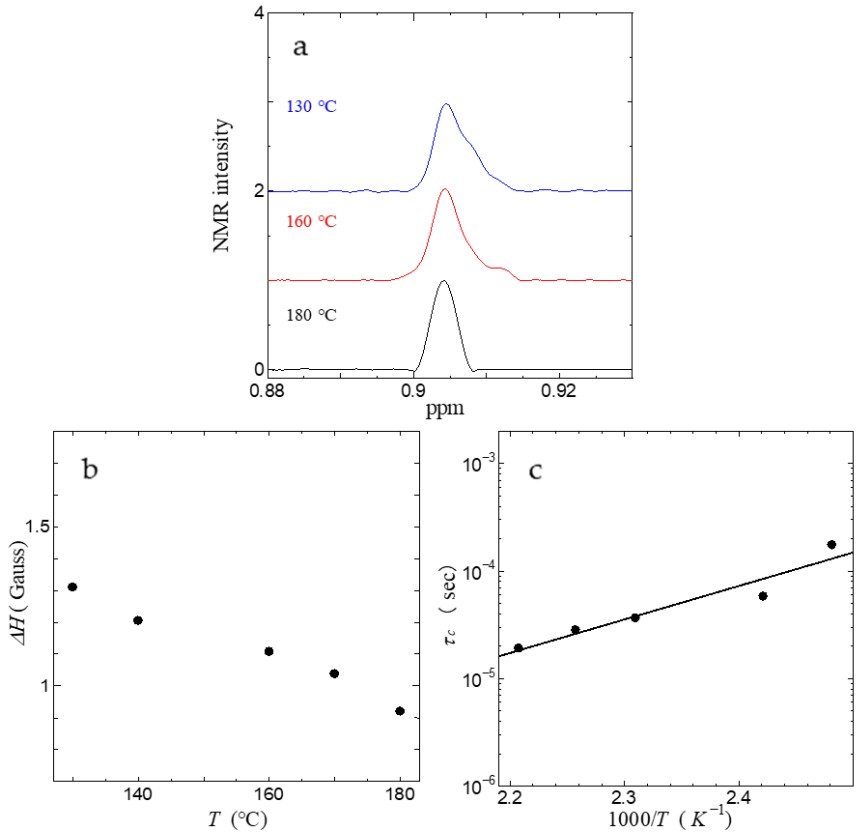

**Figure 7.** [1]H-NMR analysis of HAp-collagen composite films. (**a**) 1H-NMR lines of the HAp-collagen composite film at various temperatures. (**b**) The relationship between the second moment of the NMR spectra of composite film and temperature. (**c**) The relationship between correlation time and temperature of protons in the composite film.

Figure 7b shows the temperature dependence of the full width at half maximum ($\Delta H$) of the NMR absorption line. As shown in Figure 7b, the line width monotonously decreases with increasing temperature [40]. We can obtain the correlation time of proton motion using this temperature dependence. The proton correlation time $\tau$ of the HAp-collagen composite film was then obtained using the following Equation [41]:

$$\tau_c = \frac{2\pi}{\alpha \gamma \Delta H} tan \left[ \frac{\pi}{2} \frac{\Delta H^2 - B^2}{C^2 - B^2} \right] \qquad (2)$$

Here $\alpha$ is $(8\ln 2)^{-1}$, and $\gamma$ is a gyromagnetic ratio. $C$ is the line width before, and $B$ is the line width after narrowing. Figure 7c shows the relationship between correlation time $\tau_c$ and temperature obtained using Equation (2). As shown in Figure 7c, the correlation time $\tau_c$ becomes shorter with increasing temperature. These results indicate that proton migration becomes active due to the increase in temperature. Furthermore, the correlation time obeys the Arrhenius equation, and therefore we can obtain the activation energy for proton motion in Figure 7c. The activation energy of proton motion in the HAp-collagen composite film was determined to be 0.55 eV. These results indicate that the electrical conductivity in the HAp-collagen composite film is caused by proton migration with an activation energy of 0.55 eV.

From the NMR measurement, we have obtained that the electrical conductivity of the HAp-collagen composite film in Figure 6b is caused by proton conduction. Therefore, we investigated the relationship between proton conductivity and the molecular dynamics arising from injecting HAp into collagen film from the viewpoint of dielectric dispersion in detail. It is known that, in tissue-derived biomaterials, non-Debye dielectric dispersion corresponding to $\alpha$ dispersion is observed [42,43]. Considering this fact, in the HAp-collagen composite film, it is necessary to consider the AC proton conductivity, including the dielectric dispersion. The AC proton conductivity, including the dielectric dispersion, is described as follows:

$$\sigma_{AC} = \sigma_0 - Im\left[\omega\varepsilon_0\varepsilon_\infty + \frac{\omega\varepsilon_0(\varepsilon_s - \varepsilon_\infty)}{1 + (j\omega\tau)^\beta}\right] = \sigma_0 + \frac{\omega\varepsilon_0(\varepsilon_s - \varepsilon_\infty)(\omega\tau)^\beta \sin\left(\frac{\pi}{2}\beta\right)}{\left(1 + (\omega\tau)^\beta \cos\left(\frac{\pi}{2}\beta\right)\right)^2 + \left((\omega\tau)^\beta \sin\left(\frac{\pi}{2}\right)\right)^2} \tag{3}$$

where $\varepsilon_\infty$ is the unrelaxed static dielectric constant and $\varepsilon_s$ is the dielectric constant. The symbols $\tau$ and $\beta$ are the relaxation time for the dielectric dispersion and the degree of multi-dispersion, respectively. We can calculate the frequency dependence of $\sigma_{AC}$ in the HAp-collagen composite film using Equation (3). The calculated result in the HAp-collagen composite film is shown as the solid line in Figure 7. The values of the parameter used in the fitting are listed in Table 1. As shown by the solid line in Figure 6b, it is evident that the calculated result is in good agreement with the experimental one. These results indicate that new dielectric dispersion appears by introducing HAp into the collagen. Figure 8a shows the temperature dependence of DC proton conductivity $\sigma_0$ in the HAp-collagen composite film. As shown in Figure 8a, log $\sigma_0$ increases with increasing temperature and is proportional to $1/T$ above 130 °C. Therefore, we can obtain the activation energy from the temperature dependence of $\sigma_0$ in Figure 8a. From this, the activation energy for proton conductivity is found to be 0.56 eV in the HAp-collagen composite film. This value is in good agreement with the activation energy of 0.55 eV obtained from proton correlation time in $^1$H-NMR measurement. In this way, in the HAp-collagen composite film, a new proton pathway appears by introducing HAp to collagen, and proton conductivity appears by proton motion with an activation energy of 0.55 eV.

**Table 1.** Values of $\sigma_0$, $\varepsilon_s$-$\varepsilon_\infty$, $\tau$, and $\beta$ used in the fitting of the observed frequency dependence of $\sigma_{AC}$ in the HAp-collagen composite film.

| $T$ (°C) | $\sigma_0$ (S/m) | $\varepsilon_s$-$\varepsilon_\infty$ | $\tau$ (s) | $\beta$ |
|---|---|---|---|---|
| 100 | $1.2 \times 10^{-8}$ | 4.9 | $1.8 \times 10^{-3}$ | 0.27 |
| 140 | $6.7 \times 10^{-8}$ | 6.6 | $1.3 \times 10^{-3}$ | 0.30 |
| 170 | $2.0 \times 10^{-7}$ | 8.0 | $2.0 \times 10^{-4}$ | 0.44 |
| 200 | $4.8 \times 10^{-7}$ | 17 | $4.8 \times 10^{-4}$ | 0.45 |

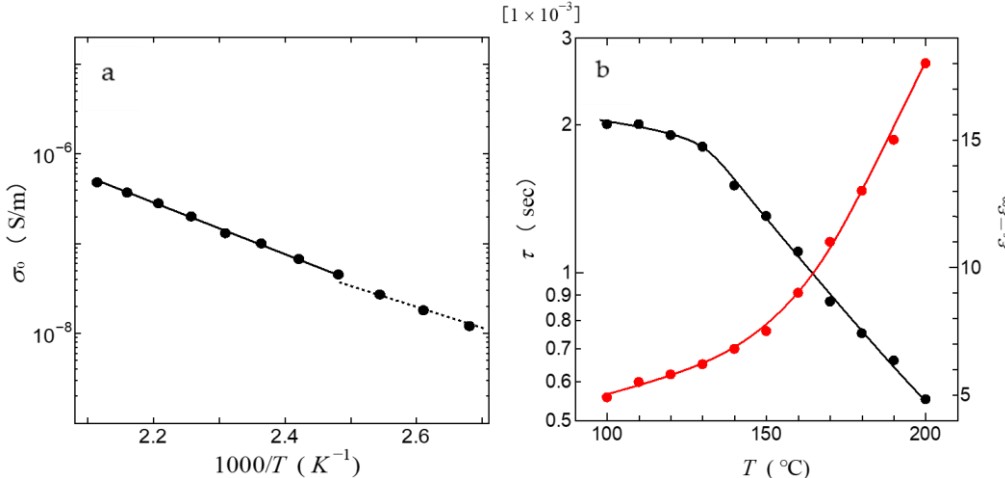

**Figure 8.** Date fitting of the HAp-collagen composite film. (**a**) Arrhenius plot of proton conductivity in the HAp-collagen composite film. (**b**) Temperature dependence of $\varepsilon_s$-$\varepsilon_\infty$ and $\tau$ in the HAp-collagen composite film.

Figure 8b shows the temperature dependence of the dielectric constant $\varepsilon_s$-$\varepsilon_\infty$ in the HAp-collagen composite film obtained from Equation (2). As shown in Figure 8b, $\varepsilon_s$-$\varepsilon_\infty$ monotonously increases with increasing temperature. This result indicates that the component of dielectric dispersion increases with increasing temperature. Considering that the increase in the electric dispersion occurs simultaneously with the increase in proton conductivity, the dielectric dispersion from introducing HAp is closely related to the flip-flop motion of OH groups which includes protons.

The dynamics concerning the flip-flop motion of OH groups can be discussed from the behavior of the dielectric relaxation time [7,8]. As shown in Figure 8b, $\tau$ increases with increasing temperature. It is also noted that the value of $\tau$ is as long as $10^{-7}$ s or more. Considering that the dielectric relaxation time of free water is less than $10^{-9}$ s, it is deduced that the observed dielectric relaxation results from the bound OH groups bonded with the collagen main-chain are similar to those observed in water bridges of humidified DNA [7,8]. That is, the flip-flop of OH groups is closely related to the increase in proton conductivity. It is noted that the proton correlation time obtained from the NMR measurement is long compared with the dielectric relaxation time observed from dielectric dispersion. Considering that the proton correlation time means that the long-range proton migration time is directly related to the proton conductivity, and the flip-flop motion of OH groups is short-range dynamics, these results are reasonable. In addition, it is speculated that the flip-flop motion of OH groups is closely related to the appearance of proton migration.

In order to obtain information on the formation of the proton-conduction pathway, we synthesized HAp-collagen composite films of various HAp contents and carried out the FT-IR and proton-conductivity measurements. Figure 9a shows the FT-IR spectra of the HAp-collagen composite films synthesized with various HAp contents. As shown in Figure 9a, several strong absorption peaks are observed. As described above, the absorption peaks at 1240 cm$^{-1}$, 1480 cm$^{-1}$, 1540 cm$^{-1}$, and 1650 cm$^{-1}$ are derived from the collagen. On the other hand, the absorption peaks at 560 cm$^{-1}$, 600 cm$^{-1}$, and 1030 cm$^{-1}$ are derived from the P-O bond in the HAp, and these peaks become strong with the increase in the HAp contents, as shown in Figure 9a. Using the weight ratio of HAp to collagen, the HAp content $n$ (the number of HAp per collagen peptide representing Gly-Pro-Hyp) can be determined for values of 0, 5, 15, 25, 31, and 40% HAp content as corresponding to values of 0, 0.029, 0.11, 0.19, 0.25, and 0.38.

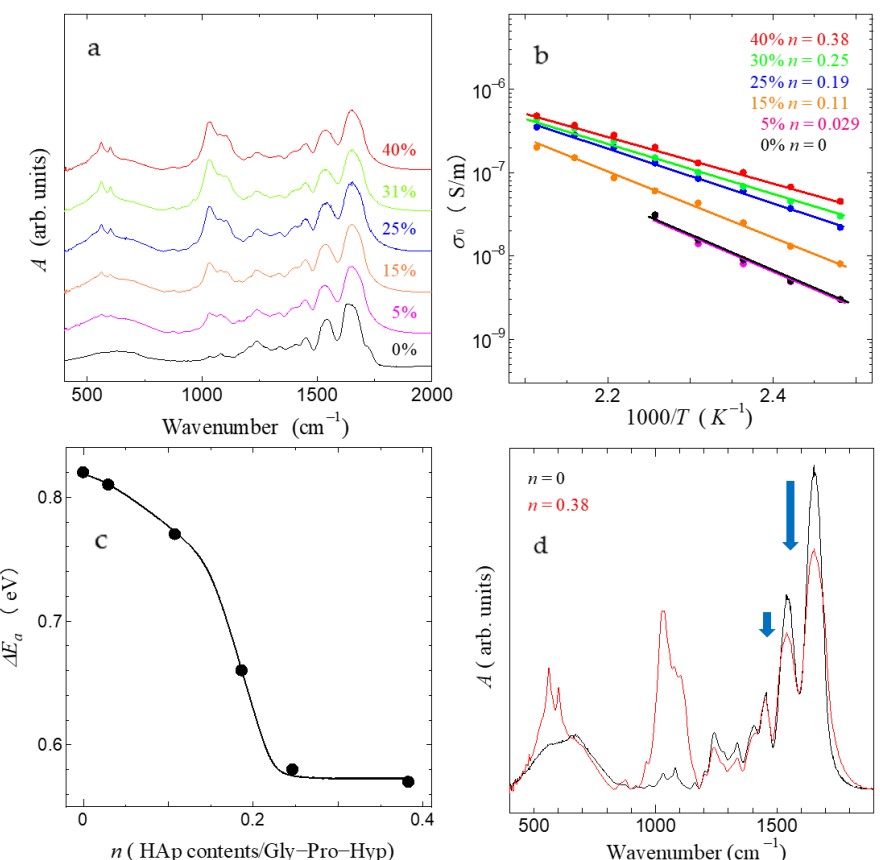

**Figure 9.** Proton conduction properties of HAp-collagen composite films with different HAp contents. (**a**) FT-IR spectra of HAp-collagen composite for each HAp weight ratio, (**b**) the relationship between proton conductivity and temperature for the HAp-collagen composite films at different concentrations, (**c**) the relationship between $\Delta Ea$ and HAp content $n$ (**d**) FT-IR spectra of collagen film and the HAp-collagen composite film of HAp weight ratio 40%.

Figure 9b shows the temperature dependence of proton conductivity in the HAp-collagen composite film at various HAp contents. Here, the DC proton conductivity $\sigma_0$ in the HAp-collagen composite films was obtained by fitting the measured frequency dependence of AC conductivity with Equation (1). As shown in Figure 9b, the DC proton conductivity in the HAp-collagen composite films increases by increasing $n$, although DC conductivity in the specimens of $n < 0.11$ cannot be obtained by melting above 160 °C. In addition, it is noted that DC proton conductivity obeys the Arrhenius equation. Therefore, we can estimate the activation energy from Figure 9b.

Figure 9c shows the HAp content $n$ dependence of the activation energy $\Delta E_a$ in the HAp-collagen composite film. As shown in Figure 9c, $\Delta E_a$ in the HAp-collagen composite films gradually decreases until $n\sim0.1$, drastically decreases at around $n\sim0.2$, and becomes almost constant above 0.25. This result indicates that the proton-conduction pathway in the HAp-collagen composite film is newly formed by injecting the HAp into collagen film above $n\sim0.25$. Here, we focus on the IR spectra of the HAp-collagen composite film and collagen in order to consider the bonding between the HAp and collagen. Figure 9d shows IR spectra of the collagen and the HAp-collagen composite film of the HAp weight ratio of 40%. As shown in Figure 9d, the peak intensities at 1650 cm$^{-1}$ and 1540 cm$^{-1}$ in the HAp-collagen composite film with a HAp content of $n = 0.38$ were decreased. As described above, in the HAp-collagen composite film, carbonyl oxygen and amide nitrogen in collagen bond with divalent metal ions. Furthermore, the bands of amide I and amide II at 1650 cm$^{-1}$ and 1540 cm$^{-1}$ are absorption peaks originating from C=O stretching vibration and N-H inflection vibration and; C-N stretching motion, respectively. Considering these

facts, it is deduced that the decreases in the peaks at 1650 cm$^{-1}$ and 1540 cm$^{-1}$ result from the binding of HAp to collagen, which inhibits the vibrational motion of amide I and amide II [34]. In this way, the injection of HAp to collagen yields the binding of calcium in HAp to the carbonyl oxygen and amide nitrogen in collagen. It is known that the HAp-collagen composite film is produced in vivo by the synthesis of HAp on a collagen scaffold. Carboxyl oxygen and amide nitrogen in collagen bond with divalent metal ions such as calcium ions and HAp grows along collagen fibers, chelated to carboxyl oxygen and the amide nitrogen [34]. That is, the HAp is synthesized on a collagen-fiber scaffold, and the HAp-collagen composite film has a structure in which the collagen fiber and the HAp *c*-axis are aligned in parallel.

From these results, we can deduce the proton conduction model in the HAp-collagen composite film, as shown in Figure 10. It is reported that the proton-conduction pathway in HAp at high temperature is formed between OH groups (dotted line in Figure 10) [23]. It is also known that glycine, proline, and hydroxyproline (Gly-Pro-Hyp) make up the majority of the amino acid sequence that constitutes collagen. Recently, it has been shown that proton conduction in the hydrated collagen occurs in the Gly-Pro-Hyp sites. This work suggests that the hydroxyl groups of Gly-Pro-Hyp and HAp, which form the majority of the collagen in the HAp-collagen composite film, also cause proton conduction. In the HAp-collagen composite film, as shown in Figure 10, new hydrogen bondings are formed between the HAp and collagen, and the connecting of hydrogen bonds appears as a solid green line in Figure 10. This means that the proton-conduction pathway appears by the breaking of the hydrogen bond. The OH groups supply protons, and carboxyl oxygen groups behave as the proton acceptor. That is, protons move in the HAp-collagen composite film via the new bonding formed between HAp and OH and carboxyl oxygen groups in collagen by the breaking and rearrangement of hydrogen bonds.

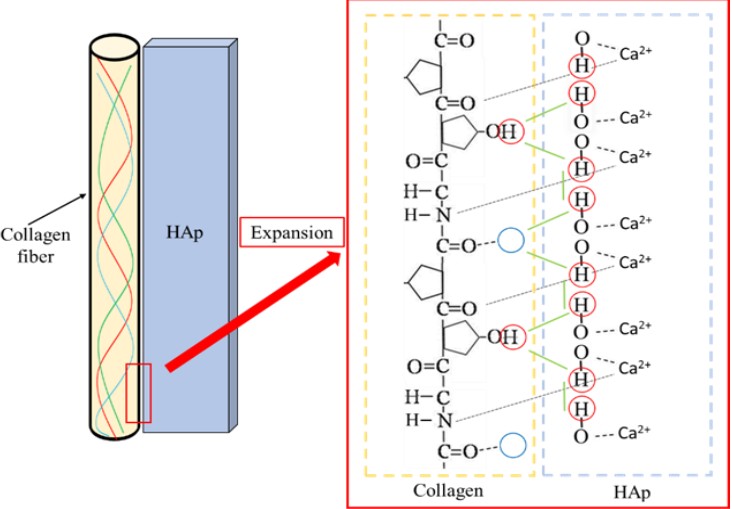

**Figure 10.** Proton-conductivity pathway in the HAp-collagen composite.

Finally, we confirm the consistency of the proton conductivity model in Figure 10. Figure 11 shows the result of the measurement of the HAp-collagen composite film in the film vertical and fiber direction. As shown in Figure 11, proton conductivity in the film vertical direction is much lower than that measured in the fiber direction. This result suggests that the proton-conduction pathway is formed along the collagen fiber direction, as shown in Figure 10. This result indicates that the proposed conducting model is consistent. However, the proton-conduction pathway perpendicular to the fiber direction is not clear in this model. The proton conductivity perpendicular to the fiber direction may be formed between collagen fibers. This result will appear in future papers.

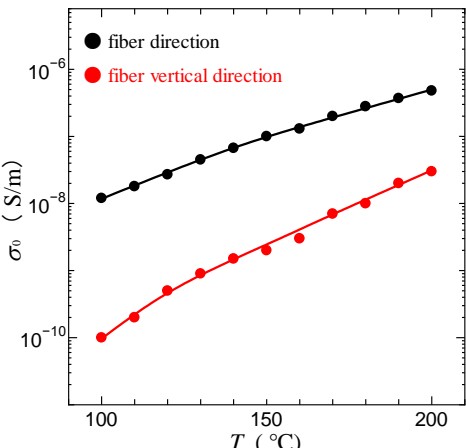

**Figure 11.** Relationship between fiber direction and conductivity of the HAp-collagen composite film.

## 4. Conclusions

In this study, proton conductivity in the HAp-collagen composite film has been investigated using FT-IR, NMR, and electrical measurements. The conductivity measurement revealed that the electrical conductivity in the HAp-collagen composite film becomes approximately ten times higher than the conductivities of the HAp film and collagen film. Furthermore, the proton NMR measurement in the HAp-collagen composite film showed a motional narrowing of the NMR absorption lines above 130 °C. Furthermore, the activation energy calculated from the proton correlation time $\tau_c$ obtained from proton NMR measurement was 0.55 eV. This activation energy is in good agreement with that of 0.56 eV obtained from the electrical conductivity measurement. These results indicate that the HAp-collagen composite film becomes a proton conductor without external humidification conditions above 130 °C. In addition, proton conductivities in various HAp contents in the HAp-collagen composite films have been measured. These results indicated that the value of the activation energy of proton conductivity was lowered, accompanied by the formation of the HAp-collagen composite film, and saturated at $n > 0.25$. From these results, the HAp-collagen composite film forms the proton-conduction pathway $n > 0.25$ and becomes the proton conductor with no external humidification under conditions of $n > 0.25$ above 130 °C.

**Author Contributions:** Conceptualization, Y.M.; methodology, T.F.; validation, T.F. and Y.M.; formal analysis, T.F. and Y.M.; investigation, T.F. and Y.M.; data curation, T.F. and Y.M.; writing—original draft, T.F.; writing—review and editing, T.F. and Y.M.; visualization, T.F. and Y.M.; supervision, Y.M.; project administration, T.F.; funding acquisition, Y.M. All authors have read and agreed to the published version of the manuscript.

**Funding:** This research received no external funding.

**Conflicts of Interest:** The authors declare no conflict of interest.

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
