# Peer review of "Anhydrous Proton Conductivity in HAp-Collagen Composite"

_jcs, doi:10.3390/jcs6080236_

Round 1

Reviewer 1 Report

Although the title is good and attracts attention, there are still many places that need to be significantly revised. These figures can be more simply integrated or provided into supplementary data.

Specific suggestions:

Abstract- line 11 “…operation above 100°C, and so on, and..”

Line 20 “proton conductivity decreases by injecting the HAp into the collagen “ These wording in many places in the abstract are inappropriate and the abstract needs to be revised to clarify the results and significance.

Materials and Method

Figure 1- please provide a scale bar and make sure the image is original.

Figure 5- please provide detailed dimensional sizes

Results and Discussion-

Lines 234-236: English needs to be revised and may be confused

Figure 6- The resolution of all FTIR figures needs to be revised, the ABCD... suggest use the functional bond marked instead

Lines 265-282 are materials and methods description, rearrangement is suggested. This section requires extensive revision. Much of this section could be moved to Materials and methods, followed by a discussion of comparisons with references.

Reviewer 2 Report

Comments:

Why do the authors keep duplicate institutional addresses as 1 and 2.

Faculty of Science & Engineering, Setsunan University, Ikeda-Nakamachi, Neyagawa 572-8508, Japan

Need to revise the abstract to mention the present study's data. Provide optimum HA and collagen content for Proton conductivity. Provide your conclusion at the end of the abstract.

proton conductivity in various HAp contents in the HAp-collagen composites: Not clear

 proton conductivity decreases by injecting the HAp into the collagen: Mention the concentration of HAp and collagen here.

 the proton NMR line is observed above 130°C: But at the end of the introduction, it was stated at 150 ÌŠC(the proton conductivity at around 150 ÌŠC)

 biopolymers such as the nucleic acid, protein, and sugar chain[6–14]. It is well-known that many types of biopolymers exist, including collagen, chitin, and cellulose: Both of these sentences give the same meaning, revise it.

 It is well-known that many types of biopolymers exist: Not a proper sentence.

proton conductivity in biopolymers [6–11,16], and proton conductivity :Reframe 

Collagen information is missing in the Introduction part. Include the information of collagen in proton conductivity

 proton conductivity at around 150 ÌŠC:Give a justification to choose 150 ÌŠC in the intro part.

The collagen membrane was prepared using the purified HAp-collagen composite: Its not clear how did you obtain collagen from the composite of HAp-collagen?

removed by FT-IR spectra: Reframe

In Figure 1, the purified collagen membrane is translucent: First, explain how did you get this membrane? fabricated or commercially bought?

The thickness of the specimen was adjusted to be 0.10 mm: How?

cut to 1.0 cm x 0.70 cm and used in measurements: Not clear, after cut why do you measure again?

Figure 2 shows micrographs of the collagen membrane and the HAp-collagen composite: Ok first explain the methods how you got these two samples? Extraction, purification and yield ?

 HAp-collagen composite at various temperatures:Mention the temperature here

the unique pattern of scale was visible up to 150 ÌŠC: From the image, there were not many differences between 150 and 160 ÌŠC. 

Why did you choose 25, 150 and 160  ÌŠC?

Figure 2. : very poor image, not clear. Add the details of all images in Fig. legend. and scale bar in all images

Figure 3: Not clear. In the text first, explain how you extracted the collagen methods. Purity and quantification of collagen after extraction.

using purified collagen as a scaffold to introduce HAp:What is the purity of collagen?

HAp membranes: membrane or powder?

were prepared by adjusting the weight of HAp: Not clear.

 purified collagen as a scaffold: so the collagen used in this study was scaffold? had 3D porous structure?

 The purified collagen was agitated: How and purity?

The HAp-collagen composite was then prepared by mixing the calcium ion-loaded collagen: Not clear reframe

2-2. Synthesis of HAp-collagen composites: As shown in Fig.1, i think you extract HAp from HAp-collagen composites from the tilapia fish scales, but again used the purified HAp to synthesis HAp-collagen composites? Not clear...

This cycle was performed for 10~100 cycles. : So what was the total quantity of collagen and HAp used finally? mention here

The impedance measurements of the HAp membrane, the collagen membrane, and the HAp-collagen composite: HAp-collagen composite or HAp-collagen membrane?

Figure 3: It was stated the HAp-collagen composite was prepared 

Figure 5.: Here the authors used the term "film", not a consistent term throughout MS.

Figure 6. FT-IR spectra of: add the data for HAp as well here.

Add XRD pattern of HAp and HAp-collagen composites

Figure 8: Why not used 150 C here?

Try to combine Fig.6-12 in two or three Figs.

Table 1:Revise the table format

Combine Fig.13.14 and 15

Fig.18: This is too many Figs. Try to combine the appropriate Figs in one, do not list a single image as Fig. Try to combine several images as one Fig. and explain in FIg legends.

 external humidification conditions above 130°C: Contradictory to the previous statement ( the proton conductivity at around 150 ÌŠC).

Round 2

Reviewer 1 Report

The authors have successfully revised their manuscript and it’s now acceptable.

Reviewer 2 Report

The authors addressed all my comments in a detailed way and improved the content of MS significantly. Pleased to accept in its present form.